# Molecular Evolution of the *Pseudomonas aeruginosa* DNA Gyrase *gyrA* Gene

**DOI:** 10.3390/microorganisms10081660

**Published:** 2022-08-17

**Authors:** Mitsuru Sada, Hirokazu Kimura, Norika Nagasawa, Mao Akagawa, Kaori Okayama, Tatsuya Shirai, Soyoka Sunagawa, Ryusuke Kimura, Takeshi Saraya, Haruyuki Ishii, Daisuke Kurai, Takeshi Tsugawa, Atsuyoshi Nishina, Haruyoshi Tomita, Mitsuaki Okodo, Shinichiro Hirai, Akihide Ryo, Taisei Ishioka, Koichi Murakami

**Affiliations:** 1Department of Health Science, Gunma Paz University Graduate School of Health Sciences, Takasaki 370-0006, Gunma, Japan; 2Advanced Medical Science Research Center, Gunma Paz University Research Institute, Shibukawa 377-0008, Gunma, Japan; 3Department of Respiratory Medicine, Kyorin University School of Medicine, Mitaka 181-8611, Tokyo, Japan; 4Department of Bacteriology, Gunma University Graduate School of Medicine, Maebashi 371-8514, Gunma, Japan; 5Department of Infectious Disease, Kyorin University School of Medicine, Mitaka 181-8611, Tokyo, Japan; 6Department of Pediatrics, Sapporo Medical University School of Medicine, Sapporo 060-8543, Hokkaido, Japan; 7Department of Applied Chemistry, College of Science and Technology, Nihon University, Chiyoda-ku 101-0062, Tokyo, Japan; 8Department of Medical Technology, Faculty of Health Sciences, Kyorin University, Mitaka 181-8621, Tokyo, Japan; 9Infectious Disease Surveillance Center, National Institute of Infectious Diseases, Musashimurayama 162-8640, Tokyo, Japan; 10Department of Microbiology, Yokohama City University School of Medicine, Yokohama 236-0004, Kanagawa, Japan; 11Department of Agriculture, Takasaki University of Health Welfare, Takasaki 370-0033, Gunma, Japan

**Keywords:** *Pseudomonas aeruginosa*, *gyrase A* gene, selective pressure, quinolone resistance

## Abstract

DNA gyrase plays important roles in genome replication in various bacteria, including *Pseudomonas*
*aeruginosa*. The *gyrA* gene encodes the gyrase subunit A protein (GyrA). Mutations in GyrA are associated with resistance to quinolone-based antibiotics. We performed a detailed molecular evolutionary analyses of the *gyrA* gene and associated resistance to the quinolone drug, ciprofloxacin, using bioinformatics techniques. We produced an evolutionary phylogenetic tree using the Bayesian Markov Chain Monte Carlo (MCMC) method. This tree indicated that a common ancestor of the gene was present over 760 years ago, and the offspring formed multiple clusters. Quinolone drug-resistance-associated amino-acid substitutions in GyrA, including T83I and D87N, emerged after the drug was used clinically. These substitutions appeared to be positive selection sites. The molecular affinity between ciprofloxacin and the GyrA protein containing T83I and/or D87N decreased significantly compared to that between the drug and GyrA protein, with no substitutions. The rate of evolution of the gene before quinolone drugs were first used in the clinic, in 1962, was significantly lower than that after the drug was used. These results suggest that the *gyrA* gene evolved to permit the bacterium to overcome quinolone treatment.

## 1. Introduction

*Pseudomonas aeruginosa (P. aeruginosa)* belongs to the family Pseudomonadaceae and is both an environmental bacterium and a pathogen associated with various diseases in humans. The pathogen can cause fatal disease in compromised hosts, such as those with sepsis, in whom there is a fatality rate of approximately 20% [1]. This outcome is mainly due to the presence in this pathogen of lipopolysaccharides, which can induce systemic, devastating immunological responses, leading to systemic inflammatory response syndrome [2]. The pathogen also shows natural resistance against a range of antibiotics [3]. These characteristics could be responsible for the occurrence of sepsis-related refractory disease in *P. aeruginosa* infection.

Antibiotics, including piperacillin, carbapenems, quinolones, and aminoglycosides, are currently used for the treatment of *P. aeruginosa*-infection-related sepsis [3]. However, the clinical abuse of these drugs results in the development of drug resistance in *P. aeruginosa*. Several reports have shown that resistance to carbapenem, aminoglycoside, and quinolone in *P. aeruginosa* is a major infection-related treatment burden [4,5]. In particular, quinolone-drug-resistant *P. aeruginosa* increases sepsis-related treatment difficulties [6].

Quinolone drugs, including ciprofloxacin, ofloxacin, and levofloxacin, are frequently used for the treatment of infections, including those involving *P. aeruginosa* [7]. Pharmacological evidence has shown that these drugs inhibit DNA gyrase in *P. aeruginosa*, leading to bacteriostatic action [8]. Mutations of the gene produce resistance to these drugs.

The *P. aeruginosa* DNA gyrase comprises two molecular subunits: A (GyrA) and B (GyrB) [9]. GyrA cleaves and rebinds double-stranded host DNA, while GyrB has ATPase activity [10]. Quinolone drugs can bind to the GyrA protein and inhibit the catalytic effect of the protein, resulting in the inhibition of bacterial DNA replication [11]. Previous reports have indicated that a 67–106 amino-acid motif in the GyrA protein is the quinolone-resistance-determining region (QRDR motif) [12]. Previous in vitro studies also suggested that amino-acid substitutions in the QRDR motif lead to a reduction in the affinity between quinolone drugs and GyrA, leading to drug resistance [13,14]. Several studies have also indicated that mutations of the amino acid residues T83 and D87, which are elements of the QRDR motif, are crucial to the development of quinolone resistance in *Pseudomonas*
*aeruginosa* [15,16,17]. However, the details of the molecular interactions between the drug and the protein remain unknown.

Advanced bioinformatics-technology-based molecular evolutionary analyses allow us to investigate the evolution and functions of proteins [18]. Simulation studies on the docking of proteins and ligands may also allow the prediction of drug effectiveness [19]. To elucidate the mechanism of quinolone resistance in *P. aeruginosa*, we performed a detailed molecular evolutionary analyses of the *gyrA* gene and GyrA protein in *P. aeruginosa* strains collected from different parts of the world and investigated the molecular interactions between *P. aeruginosa* GyrA protein and quinolone in silico.

## 2. Materials and Methods

### 2.1. Strain selection

To analyze the molecular evolution of the *P. aeruginosa gyrA* gene, we collected full-length nucleotide sequences of gene from GenBank (https://www.ncbi.nlm.nih.gov/genbank/ (accessed on 24 September 2020)). Sequences from strains without information about the regions or years in which they were detected or isolated, ambiguous sequences, and genetic recombinants were omitted from the dataset, resulting in sequences from 2535 strains. These sequences were converted to amino-acid sequences using MEGA7 [20], and multiple alignments were performed using MAFFT Version 7 [21]. Thereafter, PAL2NAL Version 14 [22] was used to reconvert the amino-acid sequences to nucleotide sequences, and Clustal Omega [23] was used to exclude sequences with 100% identity. The final dataset contained sequences from 480 strains isolated or detected between 1882 and 2019 in 40 countries.

### 2.2. Phylogenetic-Tree Construction and Estimation of Evolutionary Rate Using the Bayesian Markov Chain Monte Carlo Method

To construct a phylogenic tree and estimate the evolutionary rate of the *gyrA* gene, we used the Bayesian Markov Chain Monte Carlo (MCMC) method in BEAST Version 2.4.8 [24]. First, the best substitution model (GTR+I+G) was determined using the jModelTest 2.1.10 program [25]. Next, using the path-sampling/stepping stone sampling method [26,27], we searched for the best of four clock models—Strict Clock, Relaxed Clock Exponential, Relaxed Clock Log Normal, and Random Local Clock—and two prior tree models: Coalescent Constant Population and Coalescent Exponential Population. The optimal dataset was estimated to be that obtained using the Strict Clock and Coalescent Constant Population models. We ran the MCMC chains for 300,000,000 steps, with sampling performed every 1000 steps. Convergence was evaluated by the effective sample size using Tracer Version 1.7.1 [28], and values greater than 200 were considered to be acceptable. Phylogenetic trees were created using Tree Annotator Version 2.4.8, which is packaged with BEAST, with the first 15% of the iterations discarded as burn-in. We used FigTree Version 1.40 (http://tree.bio.ed.ac.uk/software/figtree/ (accessed on 8 June 2021)) to illustrate and edit the phylogenetic tree. We also estimated the evolutionary rate of the *gyrA* gene using Tracer Version 1.7.1 by selecting the appropriate model, as described above.

### 2.3. Analysis of Selective Pressure

To estimate the positive and negative selection sites in the *gyrA* gene of *P. aeruginosa*, we calculated the nonsynonymous (dN) and synonymous (dS) substitution rates at each codon position using the Datamonkey web server [29]. Five methods—single-likelihood ancestor counting (SLAC), fixed-effects likelihood (FEL), internal fixed-effects likelihood (IFEL), fast unconstrained Bayesian approximation (FUBAR), and mixed-effects model of evolution (MEME)—were used to estimate positive selection pressure, and four methods—SLAC, FEL, IFEL, and FUBAR—were used to estimate negative selection pressure. The significance level was set to *p*-values of *p* < 0.05 for SLAC, FEL, IFEL, and MEME, and posterior probabilities of >0.9 for FUBAR, and both positive (dN/dS > 1) and negative (dN/dS < 1) selective pressures, including the sitting position selected by SLAC, were used as the selective pressure.

### 2.4. Bayesian Skyline Plot Analysis

We estimated the genealogical population size of the *P. aeruginosa gyrA* gene using the Bayesian skyline plot algorithm in BEAST Version 2.4.8. Appropriate substitution and clock models were selected as described above. The analysis was performed in each of the four populations: no substitutions at either T83 or D87 (333 strains), substitution only at T83I (111 strains), substitution only at D87N (29 strains), and substitutions at both T83I and D87N (seven strains). The plots were visualized with 95% highest posterior density (HPD) using Tracer Version 1.7.1.

### 2.5. Structural Modeling

We downloaded the crystal structure of the GyrA protein (PDBID:3LPX), as the template for homology modeling, from the Protein Data Bank (PDB) (https://www.rcsb.org/ (accessed on 24 November 2021)), and the three-dimensional (3D) structure of ciprofloxacin (Compound CID: 2764) for the docking simulation analysis from PubChem (https://pubchem.ncbi.nlm.nih.gov/ (accessed on 24 November 2021)). The structural models of the GyrA proteins of the representative strains from each group were constructed using the homology modeling method with MODELLER Version 9.22 [30]. The sequences used for each model were as follows: prototype (T83I and D87), GenBank NSUR01000008; T83I, GenBank NZ_JACEKO010000045; D87N, GenBank LLPE01000071; T83I and D87N, GenBank LLPQ01000001. The reliability of the structures was assessed using Ramachandran plot analysis with CooT Version 0.8.9.2 [31]. Energy minimization of the models was performed using GROMOS96 [32] implemented in the Swiss PDB Viewer Version 4.1.0 [33].

### 2.6. Docking Simulation Analysis

To elucidate the details of the molecular interactions between the *P. aeruginosa* GyrA protein and ciprofloxacin, we performed docking-simulation analysis for the protein produced by each mutation, using Autodock vina [34]. Among the 20 models created by Autodock vina, the model with the best results for binding affinity and interaction sites was selected. The model with the highest binding affinity (largest absolute value) was used for the analysis of the molecular interactions between the GyrA protein and ciprofloxacin. Pymol Version 2.3.4 [35] was used to detect the interacting residues of the docking complexes, and to generate the figures in this study.

### 2.7. Statistical Analyses

To evaluate the differences in the rates of gene evolution in the strains before and after quinolones were introduced into clinical use, Mann–Whitney U tests were used. Comparisons of the binding affinities between the groups were performed using one-way analysis of variance for normally distributed variables, and Kruskal–Wallis tests for non-normally distributed variables. We also conducted Bonferroni and Tukey tests for multiple comparisons among the groups. Any *p*-values of *p* < 0.05 were considered to be statistically significant. All statistical analyses were performed using EZR Version 1.54 [36].

## 3. Results

### 3.1. Phylogenetic Tree Construction Using the Bayesian MCMC Method

A phylogenic tree of the *P. aeruginosa gyrA* gene was constructed using the Bayesian MCMC method (Beast v.2.4.8) (Figure 1). The tree showed that an ancient common ancestor of the *gyrA* gene existed 756 years ago (95% HPD, 0–1015 years ago). A common ancestor of T83I, D87N, and double substitution strains of T83I and D87N first diverged approximately 620 (95% HPD, 0–944), 254 (95% HPD, 0–521), and 75 (95% HPD, 0–172) years ago, respectively. An ancestor of T83I diverged approximately 150–200 years ago. The *P. aeruginosa gyrA* gene phylogenetic tree of all 480 strains was divided into 11 clusters (A–K). The evolutionary rates of the gene in the strains prior to the clinical usage of quinolone, before 1962, were significantly lower than those after the drug came into common usage (*p* = 1.05 × 10^−156^).

### 3.2. Selective Pressure Analysis Using the SLAC, FEL, IFEL, FUBAR, and MEME Methods

In SLAC, FUBAR, and MEME, a positive selective pressure was estimated for amino acid 51. In all selective pressure analyses performed using SLAC, FEL, IFEL, FUBAR, and MEME, two amino-acid substitutions, such as amino acid 83 and amino acid 87, were estimated to be positive selection sites. Fifty-six negative selection sites were identified in the *gyrA* gene.

### 3.3. P. aeruginosa gyrA Gene Phylodynamics Using the Bayesian Skyline Plot Method

We analyzed the phylodynamics of the *P. aeruginosa gyrA* gene using the Bayesian skyline plot (BSP) method. Figure 2 shows the detailed parameters. The mean effective population size of T83 and D87 in the *P. aeruginosa gyrA* region increased moderately until approximately 1800, then increased sharply around 1950. The mean effective population size of T831 and D87N then showed a slight increase between 1950 and 2000 and 2000 and 2005, respectively. The double-mutant T83I–D87N did not exhibit any fluctuation.

### 3.4. Molecular Interactions between P. aeruginosa gyrA and Ciprofloxacin

Figure 3a shows the results of the docking simulation analysis, indicating that the binding sites between the prototype (T83 and D87) of *P. aeruginosa* GyrA and ciprofloxacin were D87, S111, V112, G114, and L269 (Table 1). Of these, D87 was an element in the QRDR region. The intermolecular distance between *P. aeruginosa* GyrA (T83 and D87) and ciprofloxacin was 2.1–3.3 Å (Table 1). The binding affinities between the prototype (T83 and D87) and ciprofloxacin were −7.4 to −6.3 kcal/mol (Table 2). T83 and D87 had the highest binding affinity among the four groups: T83 and D87, T83I, D87N, and T83I and D87N. Figure 3b shows that the binding sites between *P. aeruginosa* GyrA T83I and ciprofloxacin were R99, M101, E513, D516, and R519 (Table 1). Of these, R99 and M101 were located in the QRDR region. The intermolecular distance between *P. aeruginosa* GyrA (T83I) and ciprofloxacin was 2.1–3.3 Å (Table 1). The binding affinities between T83I and ciprofloxacin were −6.5 to −6.0 kcal/mol (Table 2). T83I had the minimum binding affinity among the four groups: T83 and D87, T83I, D87N, and T83I and D87N. Figure 3c shows that the binding sites between D87N and ciprofloxacin were N87, D115, A117, and Q268 (Table 1). Of these, N87 was an element in the QRDR region. The intermolecular distance between *P. aeruginosa* GyrA (D87N) and ciprofloxacin was 2.2–3.3 Å (Table 1). The binding affinities between T83I and ciprofloxacin were −7.2 to −6.1 kcal/mol (Table 2). Figure 3d shows that the binding sites between the T83I–D87N double-mutant strain and ciprofloxacin were N87, F109, D115, N116, A117, and Q268 (Table 1). Of these, N87 bound to the QRDR region. The intermolecular distance between *P. aeruginosa* GyrA (T83I and D87N) and ciprofloxacin was 2.2–3.5 Å (Table 1). The binding affinities between T83I and ciprofloxacin were −7.4 to −6.1 kcal/mol (Table 2). The difference between the prototype (T83 and D87) and T83I (*p* = 0.04) was statistically significant.

## 4. Discussion

In the present study, we conducted a detailed molecular evolutionary analysis of the *P. aeruginosa gyrA* gene. First, we established a time-scaled phylogenetic tree using the MCMC method, based on the full-length *gyrA* gene of strains collected from around the world, which showed that the common ancestor of the *gyrA* gene existed approximately 760 years ago (1264 CE). The tree then formed two major clusters (Figure 1). The phylogenetic tree also showed that the quinolone-resistant strains with the quinolone-resistance-associated T83I substitution diverged first from the common ancestor, 620 years ago. Subsequently, a common ancestor of another quinolone-resistance-associated D87N substitution emerged and diverged 254 years ago. Finally, an ancestor of the strains with double-quinolone-resistance-associated T83I and D87N substitutions emerged 75 years ago. Quinolone antibiotics, including nalidixic acid, ciprofloxacin, and levofloxacin, were developed between the 1960s and the 1990s, and were widely used in clinical settings [37,38,39]. The periods between the emergence of each drug-resistant strain and the drug-resistance-associated amino-acid substitutions were mostly compatible.

Antibiotics can produce selective pressure on bacteria [40]. Positive selection sites accompanying amino-acid substitutions might be responsible for the bacteria escaping this selective pressure [41]. Quinolone drugs can bind to the GyrA protein, leading to the inhibition of DNA replication, and resulting in the inhibition of bacterial propagation 8. Previous reports have shown that the T83 and D87 amino-acid substitutions are strongly involved in quinolone resistance in the QRDR region (residues 67–106). As mentioned above, substitutions in the *P. aeruginosa* GyrA protein, including T83I and D87N, are associated with drug resistance [42]. The present data showed these substitutions to be positive selection sites. Our docking simulation data suggested that these substitutions reflected the reduced binding energy between the quinolone drug, ciprofloxacin, and the GyrA protein. The molecular length between them was also estimated. Previous reports suggested that decreased affinity is associated with the quinolone resistance of various bacteria [43]. However, to the best of our knowledge, specific data related to these aspects are lacking. Therefore, the data regarding the detailed molecular interactions collected in this study could be first in the literature, although these are in silico data. Fifty-six negative selection sites in the form of amino-acid substitutions are present in the GyrA protein. In general, negative selection acts to eliminate harmful mutations in bacteria [44,45]. Therefore, negative selection in the GyrA protein might play a role in maintaining the long-term stability of the biological functions of the bacterium. Taken together, these results indicate that *P. aeruginosa* might acquire the ability to survive quinolone drugs [3]. We also estimated the evolutionary rates of the gene in strains collected before and after the clinical use of quinolone. The evolutionary rates in the strains collected before the first clinical use of quinolone in 1962 were significantly lower than those after this point. These results suggested that the use of quinolone accelerated the evolution of this gene.

Our study showed that the T83 and D87 amino-acid substitutions in *P. aeruginosa* GyrA are essential for quinolone resistance. Resistance does not only occur in *P. aeruginosa;* there have been several reports about the effects of mutations in GyrA on quinolone resistance in members of the Gammaproteobacteria/Bacteria [15,46,47]. The GyrA of quinolone-resistant *E. coli* has substitutions in S83 and D87, corresponding to the T83 and D87 regions of GyrA in *P. aeruginosa* [48]. These substitutions may be a common pathway to resistance against quinolone drugs in Gammaproteobacteria/Bacteria.

The affinity between GyrA and quinolone was decreased by the T83I single mutation, as it was by the T83I and D87N double mutation in this study. The quinolone resistance was not affected by the D87N single mutation. A previous study suggested that T83I substitution may be primarily responsible for quinolone resistance, and the influence of the D87N mutation is weak, based on the minimal inhibitory concentration of quinolone [17]. However, the population of bacteria carrying the T83I and D87N double mutation increased after quinolone usage. Thus, the increasing population of *P. aeruginosa* with T83I and D87N substitutions in GyA is not sufficiently explained by the change in binding affinity between protein and ligand. Further studies may be needed to explain this phenomenon.

We analyzed the *gyrA* gene phylodynamics using BSP analysis. Figure 2 shows that the population with the T83I and/or D87N substitutions in the *gyrA* gene increased after the 1960s. These results suggested that quinolone-resistant *P. aeruginosa* strains became prevalent, as they could survive the drug. The epidemiological data were also compatible with the increase in *P. aeruginosa* resistance to quinolone drugs from the 1960s to 2000 [49]. A BSP analysis of current data showed that the population of bacteria with no substitutions in the *gyrA* gene increased recently (Figure 2a), possibly because the overuse of quinolones has decreased since 2000 [50]. 

Quinolone resistance in *P. aeruginosa* might be associated with mutations in not only the *gyrA*, but also the *parC* gene (topoisomerase IV) [51]. Previous reports have suggested that strains with mutations in both genes might exhibit high resistance against quinolone drugs [14]. Therefore, the analysis of the *parC* gene could be important in future work.

## 5. Conclusions

In this study, we performed detailed molecular evolutionary analyses of the *gyrA* gene in *P. aeruginosa*. We constructed an evolutionary tree, which suggested that a common ancestor of the gene existed over 760 years ago and diverged to form multiple clusters. Quinolone-resistance-associated amino-acid substitutions, such as T83I and D87N, emerged after quinolone drugs came into clinical use, and these substitutions might be responsible for selective pressure against the drugs. Our *in silico* data suggested that these substitutions decreased the affinity between the quinolone drug, ciprofloxacin, and the GyrA protein. The evolutionary rates of the gene in the strains existing before the clinical use of quinolone in 1962 were significantly lower than those after this time. Taken together, these results indicate that most existing *P. aeruginosa* acquired survival abilities due to the selective pressure of quinolone drugs.

## Figures and Tables

**Figure 1 microorganisms-10-01660-f001:**
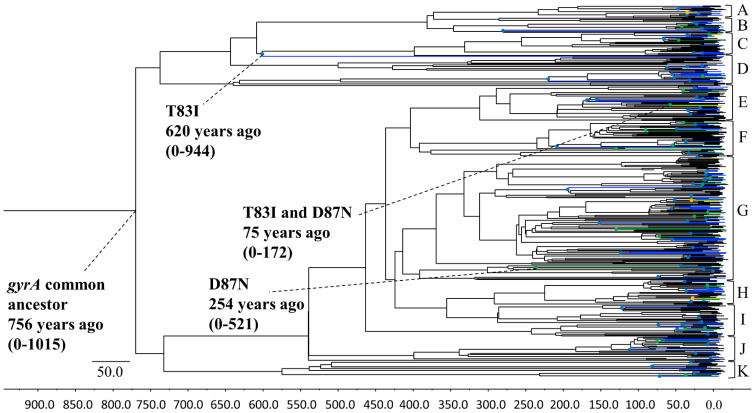
**Phylogenetic tree of *P. aeruginosa gyrA* genes constructed using the Bayesian MCMC method.** Maximum clade credibility tree from a dataset of *P. aeruginosa gyrA* genes. T83I, D87N, and double T83I-D87N mutations are shown in blue, green, and yellow, respectively. In each mutation, the first branch point (node) is represented by a plot, and the branch of the sequence is represented by the above-mentioned colors. A–K indicate 11 clusters, and parentheses indicate 95% HPDs.

**Figure 2 microorganisms-10-01660-f002:**
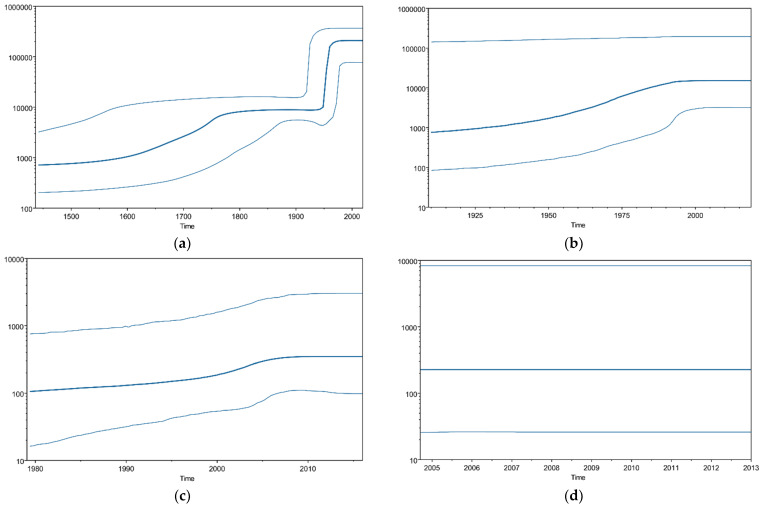
**Bayesian skyline plot for the *P. aeruginosa gyrA* genes.** Plots for (**a**) T83 and D87, (**b**) T83I, (**c**) D87N, and (**d**) T83I and D87N. The Y- and X-axes represent the effective population size on a logarithmic scale and the time in years, respectively. The solid and thin blue lines indicate the mean posterior value and the 95% HPD intervals, respectively.

**Figure 3 microorganisms-10-01660-f003:**
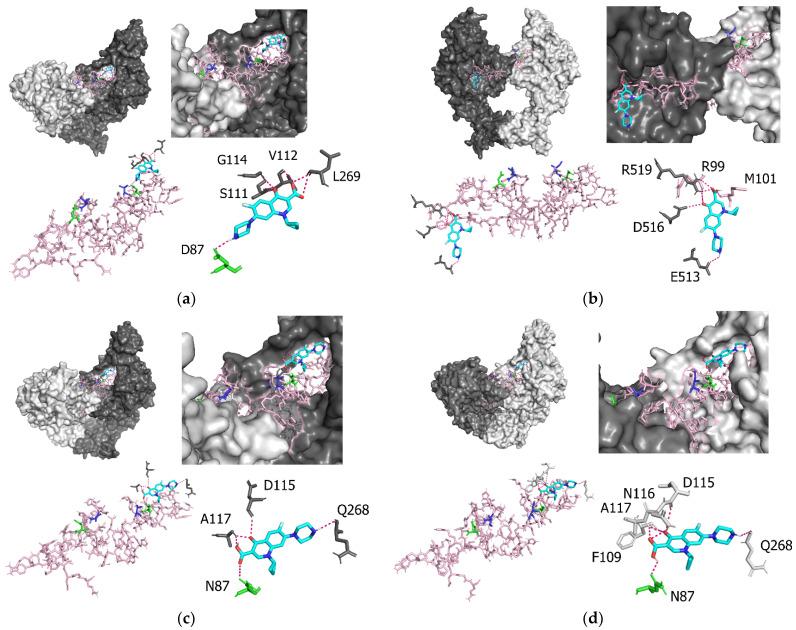
**Docking structures of the *P. aeruginosa* GyrA proteins and ciprofloxacin.** Each figure shows (**a**) structure of the *P. aeruginosa* GyrA (prototype; T83 and D87), (**b**) structure of the *P. aeruginosa* GyrA (T83I), (**c**) structure of the *P. aeruginosa* GryA (D87N), and (**d**) structure of the *P. aeruginosa* GyrA (T83I and D87N). The *P. aeruginosa* GyrA proteins are shown in gray and light gray. The QRDR region is shown in light pink. Amino-acid 83 and amino-acid 87 are indicated in blue and green, respectively. Ciprofloxacin is shown in blue.

**Table 1 microorganisms-10-01660-t001:** Intermolecular distances between *P. aeruginosa* GyrA proteins and ciprofloxacin.

Amino acid Residues of GyrA	Atoms Involved in GyrA–Ciprofloxacin Interactions	Intermolecular Distance (Å)
Prototype		
D87	O-H	2.1
S111	O-O	2.9
V112	O-H	2.3
G114	H-O	2.8
L269	O-H	2.4
L269	O-O	3.3
T83I		
R99	O-O	3.3
M101	H-O	2.2
E513	O-H	2.6
D516	O-O	3.2
R519	H-O	2.1
R519	H-O	2.5
	H-O	2.6
D87N		
N87	O-O	3.3
D115	O-O	3.3
A117	H-O	2.2
A117	H-O	2.2
Q268	O-H	2.4
D83I and D87N		
N87	O-H	2.2
F109	O-O	3.3
D115	O-O	3.5
N116	H-O	2.6
A117	H-O	2.3
A117	H-O	2.4
Q268	O-H	2.5

**Table 2 microorganisms-10-01660-t002:** Binding affinities between *P. aeruginosa* GyrA proteins and ciprofloxacin (mean ± SD).

	*P. aeruginosa* GyrA			
Mutation patterns at amino acid 83 and amino acid 87	Prototype	T83I	D87N	T83I and D87N
Molecular affinity (kcal/mol)	−6.98 ± 0.42	−6.2 ± 0.19	−6.9 ± 0.46	−6.5 ± 0.52

## Data Availability

The data presented in this study are available on request from the corresponding author.

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
