# Peer review of "Molecular Evolution of the Pseudomonas aeruginosa DNA Gyrase gyrA Gene"

_microorganisms, 2022, doi:10.3390/microorganisms10081660_

Round 1
Reviewer 1 Report
This report describes that appearance of the GryA mutant is linked to use of quinolone drugs. By in silico analyses, the authors have shown that the binding affinity between ciprofloxacin and the T83I and D87N double mutant is decreased. It seems that the paper is suitable for the aim of Antimicrobial Agents and Resistance section of Microorganisms. However, my concern is that the emergence of the T83I and D87N double mutant would not be explained (at least solely) by the reduced binding affinity to quinolone drugs, since the T83I mutant, whose emergence is not linked to use of quinolone drugs, exhibits the lowest binding affinity. This issue should be solved or at least described in the text.
Minor point:
Words shown in Figure 1 are coarse and some are difficult to identify.
Author Response
Response to Reviewer 1
  Thank you for reviewing our manuscript. According to your suggestions, we have revised the manuscript. Changes are indicated in blue. We hope that the revised manuscript is acceptable for publication in this journal.
This report describes that appearance of the GryA mutant is linked to use of quinolone drugs. By in silico analyses, the authors have shown that the binding affinity between ciprofloxacin and the T83I and D87N double mutant is decreased. It seems that the paper is suitable for the aim of Antimicrobial Agents and Resistance section of Microorganisms. However, my concern is that the emergence of the T83I and D87N double mutant would not be explained (at least solely) by the reduced binding affinity to quinolone drugs, since the T83I mutant, whose emergence is not linked to use of quinolone drugs, exhibits the lowest binding affinity. This issue should be solved or at least described in the text.
Reply 1-1
Thank you for your comments. Following your suggestion, we added some sentences describing the relationships among the T83I mutation, drug sensitivity, and the usage of quinolone drugs, and cited some new references (ref No 17, Page 6 Lines 300-309).
Minor point:
Words shown in Figure 1 are coarse and some are difficult to identify.
Reply 1-2
We have improved Figure 1 and the English usage throughout the manuscript.

Reviewer 2 Report
This is an interesting meta-analysis in the field of drug resistance. Unfortunately, the manuscript is not well written and needs editing improvement overall. In some cases, the text is not clear to support the presented conclusions. Apart from careful editing, the authors should address the points in the attached file.

Author Response
Response to Reviewer 2
  Thank you for reviewing our manuscript. According to your suggestions, we have revised the manuscript. Changes are indicated in blue. We hope that the revised manuscript is acceptable for publication in this journal.
This is an interesting meta-analysis in the field of drug resistance. Unfortunately, the manuscript is not well written and needs editing improvement overall. In some cases, the text is not clear to support the presented conclusions. Apart from careful editing, the authors should address the following points:
- Could the authors provide evidence (from the available references-data bases) on the measured in vitro activity of the different gyrase A mutants? If not, we believe that the authors should add a table where they present the different mutations of gyrase A in relation to quinolone resistance and survival.
Reply 2-1
Thank you for your comments. Previous reports have indicated that the T83I mutation is a major cause of quinolone resistance in vitro, and that the T83I and D87N substitutions are a major cause of quinolone resistance. We therefore focused on the T83I and D87N mutations. To explain this background, we added several references (Page 2 Lines 73-78).
- The authors limit their study exclusively to Pseudomonas aeruginosa. We believe that the authors should include some evidence on mutations accumulating on gyrase A from other members of Gammaproteobacteria/Bacteria due to the use of quinolones. These findings should be added to their Discussion and will further validate their specific findings on Pseudomonas aeruginosa as a more general response of bacteria to quinolones.
Reply 2-2
Thank you for your comments. As per your suggestion, we added a brief discussion of the findings in other Gammaproteobacteria, including E. coli., Klebsiella pneumoniae, and Proteus mirabilis, and added new references (Page 5 Lines 293-299).
- Docking simulations. You do not present the greater picture in an understandable way. Make a table where you compare the binding to ciproflaxin in the wt enzyme and the mutants. The way you show it now (table 2) suggests that the mutants have better binding to ciproflaxin. Why would a tighter binding of GyrA to a drug be desirable? Please explain.
Reply 2-3
Thank you for the comments. The large absolute value of affinity shows indicated binding in the docking study. To better understand the relationships between affinity and drug effectiveness, we added some text, as you suggested (Page 2 Lines 81-82).
- Presentation
Significant editing of the English language is needed. For example, in the Abstract: “Mutations in the GyrA are associated with therapeutic antibiotics resistance such as quinolone drug resistance”. Improve to “Mutations in GyrA are associated with resistance to quinolone-based antibiotics”. This sentence is followed by “From the background…”. What do you mean?
“These results suggested that the gyrA gene evolved to survive quinolone drug with the evolutionary rate alterations.”. Please change to “These results suggest that the gyrA gene evolved to permit the bacterium to overcome quinolone treatment“.
Reply 2-4
The entire manuscript has been proofread by an experienced scientific editor who is a native English speaker, through Enago (https://www.enago.com/).
Other
Improve quality of Fig. 1.
Reply 2-5
As suggested, we improved the quality of Fig. 1.
Replace “aa” with “residues” (or “amino acids”). The aa abbreviation is confusing, not a part of nomenclature and thus should be avoided.
Reply 2-6
We have amended the manuscript according to your suggestion, and have used “amino acid” throughout.
Use italics when referring to the genus Pseudomonas or the species Pseudomonas aeruginosa or
- aeruginosa throughout the manuscript. The same applies for all species and genera mentioned.
Genes are written in italics and small capitals, proteins with a first capital and the rest small letters. It is GyrA (protein) and gyrA (gene). Correct throughout the manuscript for all proteins and genes mentioned.
Reply 2-7
Thank you for bringing this omission to our attention. We have ensured that standard practice with respect to the italicization of scientific names has been followed throughout the manuscript.
Table 1. Write the distance units between amino acids.
Reply 2-8
We have added the appropriate units.
Table 1, title: “Amino acid residues of gyrAgyrA–ciprofloxacin Intermolecular distance” What do you mean? Please correct accordingly.
Reply 2-9
We have clarified this phrase.
“Table 2. Binding affinities between P. aeruginosa gyrA and ciprofloxacin (mean ± SD)”. Do you mean GyrA (the protein) and not gyrA (the gene)?
Reply 2-10
We have corrected this usage.
In table 2 “Mutation Patterns”. This is not clear. To start with, write the full replacement mutation e.g., not T83 but T83I
Reply 2-11
We have made this change throughout the manuscript.
